Co-culturing Hyphomicrobium nitrativorans strain NL23 and Methylophaga nitratireducenticrescens strain JAM1 allows sustainable denitrifying activities under marine conditions

Cucaita Alexandra
Piochon Marianne
Villemur Richard richard.villemur@inrs.ca
Centre Armand-Frappier Santé Biotechnologie, Institut National de la Recherche Scientifique , Laval , QC , Canada
Silva Pedro
Electronic publication date: 2021 Nov 1
Publication date: 2021
Volume: 9
Electronic Location ID: e12424
Received 2021 Jul 16; Accepted 2021 Oct 11
Copyright: ©2021 Cucaita et al.
Copyright year: 2021
Copyright holder: Cucaita et al.
License: This is an open access article distributed under the terms of the Creative Commons Attribution License, which permits unrestricted use, distribution, reproduction and adaptation in any medium and for any purpose provided that it is properly attributed. For attribution, the original author(s), title, publication source (PeerJ) and either DOI or URL of the article must be cited.
License URL: https://creativecommons.org/licenses/by/4.0/

Keywords: Denitrification, Methylotrophy, Biofilm, Co-culture, Hyphomicrobium, Methylophaga, Marine conditions

Funding: Natural Sciences and Engineering Research Council of Canada RGPIN-2016-06061 This research was supported by a grant to Richard Villemur from the Natural Sciences and Engineering Research Council of Canada # RGPIN-2016-06061. The funders had no role in study design, data collection and analysis, decision to publish, or preparation of the manuscript.

==============================
Background

Hyphomicrobium nitrativorans strain NL23 and Methylophaga nitratireducenticrescens strain JAM1 are the principal bacteria involved in the denitrifying activities of a methanol-fed, fluidized-bed marine denitrification system. Strain NL23 possesses the complete denitrification pathway, but cannot grow under marine conditions in pure cultures. Strain JAM1 is a marine bacterium that lacks genes encoding a dissimilatory nitrite (NO2−) reductase and therefore cannot reduce NO2−. Here, we report the characterization of some of their physiological traits that could influence their co-habitation. We also perform co-cultures to assess the potential synergy between the two strains under marine and denitrifying conditions.

Methodology

Anoxic planktonic pure cultures of both strains were grown with different concentrations of nitrate (NO3−). Anoxic planktonic co-cultures could only be cultured on low NaCl concentrations for strain NL23 to grow. Biofilm co-cultures were achieved in a 500-mL bioreactor, and operated under denitrifying conditions with increasing concentrations of NaCl. NO3− and NO2− concentrations and the protein content were measured to derive the denitrification rates. The concentrations of both strains in co-cultures were determined by quantitative PCR (qPCR). Ectoine concentration was measured by mass spectrometry in the biofilm co-culture. The biofilm was visualized by fluorescence in situ hybridization. Reverse-transcription-qPCR and RNA-seq approaches were used to assess changes in the expression profiles of genes involved in the nitrogen pathways in the biofilm cultures.

Results

Planktonic pure cultures of strain JAM1 had a readiness to reduce NO3− with no lag phase for growth in contrast to pure cultures of strain NL23, which had a 2-3 days lag phase before NO3− starts to be consumed and growth to occur. Compared to strain NL23, strain JAM1 has a higher µmax for growth and higher specific NO3− reduction rates. Denitrification rates were twice higher in the planktonic co-cultures than those measured in strain NL23 pure cultures. The biofilm co-cultures showed sustained denitrifying activities and surface colonization by both strains under marine conditions. Increase in ectoine concentrations was observed in the biofilm co-culture with the increase of NaCl concentrations. Changes in the relative transcript levels were observed in the biofilm culture with genes encoding NapA and NapGH in strain NL23. The type of medium had a great impact on the expression of genes involved in the N-assimilation pathways in both strains.

Conclusions

These results illustrate the capacity of both strains to act together in performing sustainable denitrifying activities under marine conditions. Although strain JAM1 did not contribute in better specific denitrifying activities in the biofilm co-cultures, its presence helped strain NL23 to acclimate to medium with NaCl concentrations >1.0%.

Introduction

Denitrification takes place in bacterial cells where N oxides serve as terminal electron acceptors instead of oxygen (O2) for energy production when O2 depletion occurs, leading to the production of gaseous nitrogen (N2). Four sequential reactions are required for the reduction of nitrate (NO3−) to N2, via nitrite (NO2−), nitric oxide (NO) and nitrous oxide (N2O), and each of these reactions is catalyzed by different enzymes, namely NO3− reductases (Nar or Nap), NO2− reductases (Nir), NO reductases (Nor) and N2O reductases (Nos) (Einsle & Kroneck, 2004).

This study focuses on a biofilm derived from a continuous, fluidized-bed methanol-fed denitrification reactor that treated a 3 million-L seawater aquarium containing fish, birds and invertebrates. The fluidized carriers in the denitrification reactor were colonized by naturally occurring multispecies bacteria to generate a marine methylotrophic denitrifying biofilm (Labbé et al., 2003; Villemur et al., 2019), among which the methylotrophic bacteria Methylophaga spp. and Hyphomicrobium spp. accounted for 60 to 80% of the bacterial community (Labbé et al., 2007). Two bacterial strains representative of Methylophaga spp. and Hyphomicrobium spp. were isolated from the denitrifying biofilm. They are the main bacteria responsible of the denitrification in the biofilm. Methylophaga nitratireducenticrescens strain JAM1 is capable of growing in pure cultures under anoxic conditions by reducing NO3− to NO2−, which accumulates in the medium (Auclair et al., 2010; Villeneuve et al., 2013). It was later shown to be able to reduce NO and N2O to N2 (Mauffrey et al., 2017). These activities concur with the presence of gene clusters encoding two Nar reductases, two Nor reductases and one Nos reductase (Geoffroy et al., 2018; Villeneuve et al., 2012). A dissimilatory NO-forming NO2− reductase gene (nirS or nirK) is absent, which correlates with accumulation of NO2− in the culture medium during NO3− reduction. Hyphomicrobium nitrativorans strain NL23 is capable of complete denitrification from NO3− to N2, and possesses operons that encode for the four different nitrogen oxide reductases, among which is a periplasmic Nap-type NO3− reductase. However, in pure cultures, strain NL23 cannot sustain growth under marine conditions (Martineau, Mauffrey & Villemur, 2015; Martineau et al., 2013a; Martineau et al., 2013b).

Hyphomicrobium spp. are methylotrophic bacteria that are ubiquitous in the environment (Gliesche, Fesefeld & Hirsch, 2005). They have also been found in significant levels in several methanol-fed denitrification systems treating municipal or industrial wastewaters or a seawater aquarium, and they occurred often with other denitrifying bacteria such as Paracoccus spp., Methylophilales or Methyloversatilis spp. Their presence correlates with optimal performance of bioprocess denitrifying activities (Baytshtok et al., 2008; CatalanSakairi, Wang & Matsumura, 1997; Ginige et al., 2004; Layton et al., 2000; Lemmer et al., 1997; Rissanen et al., 2017; Wang et al., 2014). Methylophaga spp. are methylotrophic bacteria isolated from saline environments (Boden, 2019; Bowman, 2005). They have been found in association with diatoms, phytoplankton blooms and marine algae, which are known to generate C1 carbons (Bertrand et al., 2015; Landa et al., 2018; Li et al., 2007).

Co-occurrence of Methylophaga spp. and Hyphomicrobium spp. has been shown in two other methanol-fed denitrification systems treating saline effluents (Osaka et al., 2008; Rissanen et al., 2016). Therefore, understanding how these two taxa collaborate could benefit in optimizing denitrification systems treating saline/brackish waters. In the present report, we compared some physiological traits of both strains that can influence their co-habitation. We then used co-culture approach to characterize the potential synergy between these two strains to achieve denitrification. Our results showed that when both strains are present, sustainable denitrifying activities were achieved and that strain JAM1 helped to acclimate strain NL23 to marine conditions.

Materials & Methods

Determination of NO3−, NO2− and biomass

NO3− and NO2− were measured either by ionic chromatography as described by Mauffrey et al. (2017) or by colorimetry based on the method described by Schnetger & Lehners (2014) (File S1). In the colorimetric method, NO3− was reduced to NO2− with vanadium (III) chloride (VCl3). NO2− concentrations were then measured with the Griess reagents. In parallel, NO2− concentrations already present in the medium were measured separately with the Griess reagents. The concentrations were determined with standard solutions. Linear response ranged from 0.1 to 2 mg-N/L (either with NO3− or NO2−). Results from the NO3− reduction to NO2− by VCl3 generated the NOx concentrations (NO3− + NO2−). NO3− concentrations were then calculated as NOx –NO2−.

The measurement of the protein content was carried out to determine the amount of biomass in our cultures or biofilm. One mL of suspended cultures was centrifuged at 16,000 g 1 min. The pellet was vortexed in 0.1 M NaOH, incubated 30 min at 70 °C, vortexed again and incubated another 30 min at 70 °C. For the biofilm, one to five supports were added and vortexed in three mL of 0.1M NaOH, and incubated as described before. The protein concentration in the biomass extracts was determined by the Quick StartTM Bradford Protein Assay (BioRad, Mississauga, ON, Canada) with bovine albumin serum as standard.

The denitrification rate was defined as the reduction rate of the NOx, assuming that NO and N2O were readily reduced (no or very low transient accumulation) as shown by Mauffrey et al. (2017) and Martineau, Mauffrey & Villemur (2015). The growth rates, the NO3− reduction rates and the denitrification rates were calculated by regression of the linear portion (the R2 coefficients of determination in most cases > 0.9) of the culture growth (OD600nm), of the NO3− concentrations and of the NOx concentrations, respectively, over time for each replicate (OD h−1; NO3− mM h−1, NOx mM h−1). The specific NO3− reduction rates and the specific denitrification rates were reported as the NO3− reduction rates and the denitrification rates divided by the quantity of biomass (mg protein or OD600nm), respectively, in a vial or in the reactor.

Culture media

The Instant Ocean (IO) seawater medium was bought from Aquarium systems (Mentor, OH, USA) and dissolved at 30 g/L. NaNO3 was added at 21.4 mM (final concentration). The medium was adjusted at pH 7.5 and autoclaved. One ml per liter of autoclaved trace metal solution (per liter: 0.9 g FeSO4.7H2O, 0.03 g CuSO4.5H2O, 0.234 g MnCl2.4H2O and 0.363 g Na2MoO4.2H2O) was added. Three mL of 0.2 µm-filtered methanol were added per liter of medium.

Planktonic pure cultures of H. nitrativorans strain NL23T (ATCC BAA-2476) were performed in the 337a medium (per liter: 1.3 g KH2PO4, 1.13 g Na2HPO4, 0.5 g (NH4)2SO4, 0.2 g MgSO4.7H2O). NaNO3, if needed, was added at 21.4 mM (final concentration). The medium was adjusted at pH 7.5 and autoclaved. These 0.2 µm-filtered solutions were added per liter: five mL trace element solution (per liter: 309 mg CaCl2.2H2O, 200 mg FeSO4.7H2O, 100 mg Na2MoO4.2H2O, 67 mg MnSO4.H2O), three mL methanol and one mL vitamin B12 (0.1 mg/mL).

Planktonic pure cultures of M. nitratireducenticrescens strain JAM1T (ATCC BAA 2433) were performed in the Methylophaga 1403 medium (per liter: 24 g NaCl, 3 g MgCl2. 6H2O, 2 g MgSO4.7H2O, 0.5 g KCl, 1 g CaCl2, 0.5 g Bis-tris, and NaNO3 with final concentrations ranging from 0 to 21.4 mM). The medium was autoclaved, and these 0.2 µm-filtered solutions were added per liter: three mL methanol, 20 mL solution T (per 100 mL: 0.7 g KH2PO4, 10 g NH4Cl, 10 g Bis-tris, 0.3 g citrate ferric ammonium, pH 8.0), one mL vitamin B12 (0.1 mg/mL), 10 ml Wolf solution [per liter: 0.5 g EDTA, 3.0 g MgSO4.7H2O, 0.5 g MnSO4.H2O, 1.0 g NaCl, 0.1 g FeSO4.7H2O, 0.1 g Co(NO3)2.6H2O, 0.1 g CaCl2 (anhydrous), 0.1 g ZnSO4.7H2O, 0.010 g CuSO4.5H2O, 0.010 g AlK(SO4)2 (anhydrous), 0.010 g H3BO3, 0.010 g Na2MoO4.2H2O, 0.001 g Na2SeO3 (anhydrous), 0.010 g Na2WO4.2H2O, and 0.020 g NiCl2.6H2O, pH 8.0]. For some assays, the NaCl concentration was adjusted in this medium as needed (from 0% to 2.75%). For solid media, bacteriological agar (1.5% final concentration; Alpha Biosciences, Baltimore MD USA) was added before sterilization.

Planktonic pure cultures and co-cultures

A loop of frozen stock of strain JAM1 (Villeneuve et al., 2013) and strain NL23 (Martineau et al., 2013b) was spread onto Methylophaga 1403 or 337a agar plates supplemented with methanol and incubated for 2–7 days. The pre-cultures were derived by taking one to three colonies to inoculate the corresponding culture medium (Methylophaga 1403 and 337a) that were then incubated for 1 to 7 days under oxic conditions without NO3− (unless specified) and incubated at 30 °C at 150 rpm. The cells were then centrifuged and dispersed in the medium before use. For anoxic cultures, the sterile media were distributed (60 mL) in sterile serologic vials. The vials were purged of O2 for 10 min with N2 (Praxair, Mississauga, ON, Canada) and sealed with sterile septum caps. The planktonic co-cultures (anoxic conditions, Methylophaga 1403 medium) were inoculated with pre-cultures of strain JAM1 and of strain NL23 with a JAM1/NL23 ratio of 1:10 and a final cell dispersion of 0.1 OD600nm. Cultures were incubated for 1–7 days at 30 °C. The medium was shaken before each sampling to disperse the biomass. No biofilm was apparent in the vial border.

Biofilm co-cultures in fed-batch reactor

The reactor is illustrated in Fig. S1 and the reactor operating procedures is described in the File S2. It consisted of an airtight 500-mL working volume. A flow cell consisting of a 50-mL tube was added in the recirculating circuit where two microscope slides were added to monitor cell attachment on the surface. The system was carefully washed with water and ethanol. Two hundred and sixty acid-washed Bioflow nine mm supports were added in the reactor, and filled with the proper autoclaved medium (Table 1). Throughout all the different operating conditions, the medium was recirculated with a peristaltic pump at 30 mL/min, and the reactor was run at room temperature (ca. 22 °C). To avoid high pressure building in the reactor because of the N2 generated by the denitrifying conditions, a gas trap was added to the reactor with tubing connected to a graduated cylinder filled with water. Gas production was recorded by water displacement in the graduated cylinder (Fig. S1). Monitoring gas production during the whole operating processing allowed a quick evaluation of the performance of the reactor. Samples of the suspended biomass (one mL) were taken at sampling ports and centrifuged; the supernatant was used to determine the NO3− and NO2− concentrations. The pellets of some of these samples were kept for DNA or protein (or both) extractions. From the DNA extracts, the concentrations of strain JAM1 and strain NL23 in the suspended biomass were determined by qPCR.

Table 1 Operating conditions of the reactors.

Medium	NaCl	Methanol	NO3−	Biofilm cultures	
	% w/v	% v/v	mM	Reactors 1, 2	Reactor 3	
Methylophaga 1403	0	0.3	21.4	ND	NL23	
Methylophaga 1403	0.5	0.1	7.14	Co-culture	ND	
Methylophaga 1403	0.5	0.2	14.3	Co-culture	ND	
Methylophaga 1403	0.5	0.3	21.4	Co-culture*	NL23*	
Methylophaga 1403	1.0	0.3	21.4	Co-culture	NL23	
Methylophaga 1403	2.0	0.3	21.4	Co-culture	NL23*	
Methylophaga 1403	2.75	0.3	21.4	Co-culture*	NL23	
IO + supplements		0.3	21.4	Co-culture	NL23	
IO without supplements		0.3	21.4	Co-culture*	NL23	
Notes.

Reactors were run with successive changes of medium following the order in the Table, with one exception: Reactor 1 was run with the Instant Ocean (IO) medium but without supplements first, and then supplements were reintroduced.

* Supports taken for RNA extraction in reactors 2 and 3.

ND, not done.

The same reactor vessel was used to carry out three reactors sequentially. Reactors 1 and 2 were inoculated with strain JAM1 and strain NL23 (biofilm co-cultures). Reactor 3 was inoculated with only strain NL23 (biofilm mono-culture). Results from reactor 1 allowed to adjust the conditions for reactor 2.

To initiate reactors 1 and 2 (co-cultures), inocula of strain NL23 (four 60 mL-vials) and strain JAM1 (three 60 mL-vials) were centrifuged (ca. 3 g wet weight, each), dispersed in 20 mL 0.5% NaCl Methylophaga 1403 medium, and added to the reactor filled with 500 mL of the same medium. To avoid strain JAM1 to outcompete strain NL23 for NO3−, NO3−was not added in the medium and the reactors were operated under the oxic conditions by injecting ambient air sterilely for 3 days with a peristaltic pump connected at the bottom of the reactor, after which air was no longer supplied and NO3− was added to create denitrifying conditions. For reactor 3, the inoculum of strain NL23 was added in the reactor with 0% NaCl Methylophaga 1403 medium, and the reactor was operated with no oxic cycle (no air bubbling) with NO3−. The reactors were then run for a week, with additions of NO3− when needed, after which the medium was replaced by a fresh one. Only bacteria attached to supports and vessel were kept in the reactor; the planktonic bacteria were discarded. Upon NO3− and NO2− reductions, the medium was changed with fresh one. Medium was changed 30 times in reactor 1 during the 19 weeks of operating times, 57 times in reactor 2 over 17 weeks, and 27 times in reactor 3 over 8 weeks (File S2). In all cases, the media (Table 1) were not purged of O2 with N2 and were added in the reactor in open air. Rapid O2 consumption by the bacterial cells provided conditions for denitrifying activities to occur. Indications that such conditions were met were by the reductions of NO3− and NO2− and substantial amount of gas production. When these production and reductions appeared stable (few days to weeks), the medium was replaced with a fresh one and the reactor was sampled at regular intervals for at least 24 h to measure the concentration of NO3− and NO2−. Afterwards, one to two supports were collected, frozen at −20 °C until use for the extractions of proteins, DNA and ectoine. In some of the operating conditions (Table 1), supports were collected for RNA extractions (reactors 2 and 3). To do so, fresh medium was added in the reactor. When NO3− was near completely consumed, 10 supports were collected in a glove box flushed with N2 and frozen immediately in liquid N2, and kept at −70 °C until use to extract total RNA. In all cases, the taken supports were replaced with new ones in the reactor. After these steps, we proceeded with new medium (increasing concentration of methanol/NO3− or NaCl) as described in Table 1. When the different operating runs for reactor 1 were finished, all the colonized supports were frozen at −20 °C for further analysis. The vessel was then washed thoroughly and sterilized. New supports were added to start reactor 2. The same was done with reactor 3.

Ectoine

The protocol was based on Chen et al. (2015) and Zhang, Lang & Nagata (2009). The biofilm on one support was directly extracted with 2 mL of methanol/chloroform/water (10:5:4) with intermittent strong agitation for 3 h. The extract was then centrifuged at 16,000 g for 10 min. One mL of the supernatant was extracted with one mL chloroform/water (1:1) with agitation for 30 min. The extract was then centrifuged at 16,000 g for 10 min. The water phase was collected and dried at 35 °C with N2 for 30 min. The extract was then dissolved in one mL water and filtered on 0.2 µm filter. The stable isotopically labelled internal standard 5,6,7,8-tetradeutero-4-hydroxy-2-heptylquinoline (HHQ-d4) was added in the samples (2.66 µg/mL final concentration). HHQ-d4 was provided by the laboratory of Eric Déziel (INRS, Laval, QC, Canada). Ectoine ((S)-2-methyl-1,4,5,6-tetrahydropyrimidine-4-carboxylic acid, 97% purity) was bought at Sigma Aldrich (Oakville, ON, Canada), and used to derive a standard curve.

Fifteen-µL of the samples were analysed by a HPLC-coupled to a mass spectrometer (Waters, Milford, MA, USA). The Waters 2795 HPLC system was equipped with a Kinetex (100 x 4.6 mm) 2.6-µm C8 reverse-phase LC column (Phenomenex, Torrance, CA, USA). The mobile phase was a gradient of 1% acetic acid in water (solvent A) and 1% acetic acid in acetonitrile (solvent B) programmed as follows: initial 2% solvent B (0–4 min), 2–70% solvent B (4–5 min), 100% solvent B (5–8 min), then hold 3 min and followed by 3 min of re-equilibration. HPLC flow rate was 400 µL/min split to 40 µL/min by a Valco tee splitter. The Quattro Premier XE mass spectrometer was equipped with a Z-spray interface using electro-spray ionization in positive mode (ESI-MS/MS). Multiple Reaction Monitoring (MRM) mode was used to quantify ectoine. MassLynx and QuanLynx software (Ver. 4.1) were used. The capillary voltage was set at 3.0 kV and the cone voltage at 30 V. The source temperature was kept at 120 °C. N2 was used as nebulising and drying gas at flow rates of 15 and 100 mL/min, respectively. In MRM mode, the following transitions were monitored: for ectoine 143 →98 and for the internal standard HHQ-d4 248 →163. The pressure of the collision gas (argon) was set at 2 × 10−3 mTorr and the collision energy at 30 V for all transitions. The area of each chromatographic peak was integrated and the ectoine/HHQ-D4 response ratio was used to create the calibration curve (R2 = 0.9945) and to quantify the concentration of ectoine in the samples.

Fluorescence in situ hybridization (FISH)

The biofilm that colonized the microscope slides in the flow cell chamber was fixed with 4% cold paraformaldehyde in PBS for 60 min, and washed twice with water. The biofilm was dehydrated for 3 min successively in 50%, 80% and 95% ethanol/water, and then air dried. Hybridization buffer for both probes consisted of 20% formamide, 20 mM Tris-HCl pH 8.0, 0.01% SDS and 0.9 M NaCl. The probe concentrations used were between 10 to 20 ng/µL. Hybridization was carried out in an Omnislide in situ thermal cycler (Thermo Electron Corporation, Waltham, Mass., USA) for 3 h at 46 °C, followed by 2 min at 48 °C. Slides were washed 20 min at 48 °C, in 20 mM Tris-HCl pH 8.0, 5 mM EDTA, 0.01% SDS and 210 mM NaCl. FISH samples were mounted with Prolong Gold agent containing DAPI (Molecular Probes, Fisherscientific, Ottawa, ON Canada). The probes for Methylophaga spp. (MPH-730-Atto488; 5′CAGTAATGGCCCAGTGAGTCGCC 3′) (Janvier, Regnault & Grimont, 2003) and for Hyphomicrobium spp. (Hypho-Cy3; 5′TCCGTACCGATAGGAAGATT 3′) (Juretschko et al., 2002) were synthesized by Biomers.net (Ulm, Germany) and AlphaDNA (Montreal, QC, Canada), respectively. Slides were examined on an epifluorescence Leica DM3000 microscope.

DNA and RNA extraction, qPCR and RT-qPCR, and RNA sequencing

DNA and RNA extractions of the planktonic biomass and of the biofilm from 1 to 4 supports were performed as described (Villemur et al., 2019; Mauffrey et al., 2017; Mauffrey, Martineau & Villemur, 2015). Concentrations of strain JAM1 and strain NL23 were performed by qPCR with primers targeting narG1 and napA, respectively, as described by Geoffroy et al. (2018) (Table 2). Standard curves were performed with PCR-amplified DNA fragments of narG1 and napA from strain JAM1 genome and strain NL23 genome, respectively (Table 2).

Table 2 Sequences of the oligonucleotides used for qPCR and RT-qPCR assays.

Name	Sequence 5′–3′	Th ∘C	Length bp	
Standards for qPCR				
Strain NL23				
napA-1171f	TACAACGTCCACCTGCTGAC	55	625	
napA-1846r	TCCGCTTCGTGGTTTTCGTA			
Strain JAM1				
narG1-G	ATGACAAGATCGTGCGTTCT	57	664	
narG1-D	GGTGTACGGGTCATTGGTAAG			
qPCR				
Strain NL23 (napA)				
napA-1415f	AGGACGGGCGGATCAATTTT	61	131	
napA-1526r	CGGATATGCATCGGACACGA			
Strain JAM1 (narG1)				
narG-1313f	AGCCCACATCGTATCAAGCA	61	149	
narG-1461r	CCACGCACCGCAGTATATTG			
RT qPCR				
Strain NL23				
napA-1415f	AGGACGGGCGGATCAATTTT	61	112	
napA-1526R	CGGATATGCATCGGACACGA			
nirKf	CGCACAACATCGACTTCCA	61	130	
nirKr	GCGCAGTGATAGACGAAAAC			
rpoBf	GCCATCAACAAGCAGTACGA	61	128	
rpoBr	GCCACGAAGACCTTGACCAT			
dnaGf	CCCGATCAAAACGCCAAGTA	61	141	
dnaGr	CGCATCCATGTAGCCTTCGA			
Strain JAM1				
narG1-1313f	AGCCCACATCGTATCAAGCA	60	149	
narG1-1461r	CCACGCACCGCAGTATATTG			
narG2-597f	TTACGCTGCAGGATCACGTT	60	127	
narG2-723r	TGACTCGGGTACATCGGTCT			
rpoB-3861f	TGAGATGGAGGTTTGGGCAC	60	146	
rpoB-4006r	GCATACCTGCATCCATCCGA			
dnaG-774f	CATCCTGATCGTGGAAGGTT	60	121	
dnaG-894r	GCTGCGAATCAACTGACGTA)			
Notes.

Th, hybridization temperature.

RT-qPCR were performed as described by Mauffrey et al. (2017) with primers targeting narG1 and narG2 for strain JAM1, napA and nirK for strain NL23, and the reference genes rpoB and dnaG specific for each strain (Table 2). The standard curves were performed with dilutions of the respective genomic DNA.

The RNA samples were sequenced at the Centre d’expertise et de services Génome Québec Montreal QC, (Canada) (Sequencing type: Illumina NovaSeq 6000 S4 PE100 - 25M reads; Library Type: rRNA-depleted for bacteria). The sequencing data were uploaded to the Galaxy web platform, and we used the public server at usegalaxy.org to analyze the data (Afgan et al., 2018). Raw reads were filtered to remove low quality reads using FASTX toolkit by discarding any reads with more than 10% nucleotides with a PHRED score <20. The resulting reads were aligned to the genome of M. nitratireducenticrescens strain JAM1 (GenBank accession number CP003390.3) and to the genome of H. nitrativorans strain NL23 (CP006912.1) using Bowtie2 with default parameters. BEDtools were used to assign the number of reads to the respective genes in the genome, which were then normalized as transcripts per million (TPM). Because the samples were from one reactor run, statistical analysis could not have been performed. However, an arbitrary variation of 10% in the read values revealed that the ratio of TPM of a gene from one condition to another of >2 or <0.5 is significantly different.

Results

Growth and NO3− affinity

We compared the growth pattern of M. nitratireducenticrescens strain JAM1 and H. nitrativorans strain NL23 cultured under anoxic conditions. Strain NL23 cultures showed a 48 to 72-h lag before growth occurred, whereas strain JAM1 cultures presented no such lag (Fig. 1A). Strain NL23 anoxic cultures reached higher level of biomass than strain JAM1 anoxic cultures. We then measured the NO3− affinity by both strains, which can be an important factor of competition between them for growth. Planktonic anoxic pure cultures of strain JAM1 and strain NL23 were performed with different concentrations of NO3− to derive their respective maximum growth rates (µmax) and half-saturation constants of NO3− concentration for growth (Table 3). The µmax of strain JAM1 cultures were 68% higher than those of strain NL23 cultures (Fig. 1B, Table 3). Strain NL23 cultures showed some growth inhibition at 107 mM NO3− (Fig. 1B). To assess the affinity of these strains toward NO3− for growth, the µmax/Ks ratio was calculated (Healey, 1980) (Table 3). This ratio in strain JAM1 cultures was twice higher than that of strain NL23 cultures. The NO3− reduction rates in the strain NL23 cultures increased linearly with the increase of NO3− concentrations (from 2 to 40 mM) in the medium (Fig. 1C). These rates reached a plateau at 24 mM NO3− in strain JAM1 cultures, and showed no significant changes at higher concentrations. The specific NO3− reduction rates (rates normalized by the biomass) were however, relatively constant for both strains whatever the NO3− concentrations (Fig. 1D). These specific rates were six times higher (p < 0.001) in strain JAM1 cultures (average 4.48 ± 0.96 NO3− h−1 OD−1) than in strain NL23 cultures (average 0.77 ± 0.32 NO3− h−1 OD−1). All these results suggest that strain JAM1 cells process NO3− (uptake and reduction) at a higher rate than strain NL23.

Figure 1 Growth and NO3− reduction in planktonic mono-cultures.

(A) Growth under planktonic anoxic conditions with 25 mM NO3− and 0.3% methanol at 30 °C. (triplicate cultures). (B and C) Growth rates and NO3− reduction rates, respectively, at different NO3− concentrations. The curves in (B) are derived from non-fit linear regression; the growth rates at 107 mM in the strain NL23 cultures were not included. Each point is the average of three to eight replicate anoxic cultures. (D) Specific NO3− reduction rates. These rates were calculated with the NO3− reduction rates by the generated culture biomass (OD600 nm) at the end of the exponential phase. Each point represents data from triplicate cultures. Data for strain JAM1 in C and D were taken from Geoffroy et al. (2018).

Table 3 Kinetics of growth of strains JAM1 and NL23 under anoxic conditions.

	Strain JAM1a	Strain NL23	
Lag (h)	<6	48–72	
µmax (OD600nm h−1)	0.0110 (0.0007)	0.0066 (0.0005)	
Ks (mM)	12.6 (2.1)	15.3 (3.2)	
µmax/Ks (µM−1h−1)	0.88	0.43	
Notes.

µmax maximum growth rates

Ks half-saturation constants of NO3− for growth

Values are derived from non-regression linear measurements (Fig. 1), with standard error between parentheses.

a Data from Mauffrey, Martineau & Villemur (2015) and from new measurements.

Anoxic planktonic co-cultures

Co-cultures were first attempted under planktonic conditions to assess the effect on the denitrifying activities of culturing both strains together. Initially, we wanted to perform planktonic co-cultures under marine conditions as both strains originated from a marine denitrification system. However, strain NL23 planktonic growth is impaired when the culture medium is >1% NaCl (Martineau, Mauffrey & Villemur, 2015), impeding planktonic co-cultures under marine conditions. We found that the Methylophaga culture medium (Methylophaga 1403) with 0.5% NaCl instead of 2.4% can sustain both strains with minimal growth defect. This medium was used to perform our anoxic planktonic co-culture assays. Higher levels of strain NL23 in the inoculum (JAM1/NL23 ratio of 1:10) had to be used because of the higher affinity of strain JAM1 towards NO3− as shown above. In the first co-culture attempts, both inocula were derived from oxic pre-cultures. Anoxic planktonic mono-cultures of both strains were performed in parallel also in this medium as control. Our results showed that the co-cultures could not reduce NO2− (Fig. 2A). The NO3− reduction rates were similar between the co-cultures (0.68 ± 0.01 mM-NO3− h−1) and strain JAM1 mono-cultures (0.82 ± 0.02 mM-NO3− h−1), and NO2− accumulated in the medium (Figs. 2A, 2C). The proportion of strain NL23 at the end of the co-cultures was 1.6% (Fig. 2B).

Figure 2 Dynamics of denitrification of the planktonic co-cultures.

(A and D) Strain JAM1 and strain NL23 were co-cultured in the 0.5% NaCl Methylophaga 1403 medium under anoxic conditions with 21.4 mM NO3− and 0.3% methanol at 30 ° C. As controls, mono-cultures were carried out with both strains under the same conditions (C and F). Strain NL23 inoculum was from pre-cultures cultured under oxic conditions (no NO3−; A and C) or under anoxic conditions (with NO3−; D and F). The growth (OD 600 nm) and the concentrations of NO 3− and NO2− were measured. Results are from triplicate cultures and representative of three sets of triplicate assays. (B and E) Concentrations at the end of the culture assays of strain JAM1 and strain NL23 determined by qPCR with primers targeting napA for strain NL23 and narG1 for strain JAM1. Results in (B) are from the non-stimulated NL23 co-cultures (A), and, in (E), from the stimulated NL23 co-cultures (C). Results are from triplicate cultures.

Other planktonic co-cultures were performed by inoculating strain NL23 that was derived this time from pre-cultures cultured under anoxic conditions to stimulate its denitrification pathway. These co-cultures were capable of denitrification (Fig. 2D). The NO3− reduction rates were similar between the co-cultures (1.17 ±0.05 mM-NO3− h−1) and strain JAM1 mono-cultures (1.00 ± 0.03 mM-NO3− h−1) (Figs. 2D, 2F). NO2− accumulated transiently after NO3− reduction in the co-cultures and then was completely consumed after 36 h (Fig. 2D). The denitrification rates of the co-cultures were twice higher (t-test, p = 0.0038) (0.91 ± 0.03 mM-NOx h−1) than those in strain NL23 mono-cultures (0.43 ± 0.02 mM-NOx h−1). The proportion of strain NL23 was about 12 times higher in these co-cultures than that in the first co-culture assays (19.7% vs 1.6%; Fig. 2E). The total concentrations of both strains in these co-cultures (18.8 ± 2.4 × 108 gene copies/mL) were three times higher than those in the first co-culture assays (6.1 ± 1.2 × 108 gene copies/mL), suggesting better growth for both strains.

Biofilm co-cultures

As mentioned before, planktonic co-cultures could not be performed under marine conditions without impairing strain NL23 growth. As strain JAM1 and strain NL23 were isolated from a biofilm, developing a biofilm co-culture provided a mean to assess the adaptability of strain NL23 to marine conditions. Two reactors with Bioflow nine mm supports and containing the Methylophaga 1403 medium at 0.5% NaCl were inoculated with both strains, and operated under fed-batch mode and denitrifying conditions. The reactors were run for several weeks, with several medium changes where the planktonic cells were discarded, for the biofilm to build up and the denitrifying activities to establish. The biofilm was then acclimated with increasing concentrations of NaCl (Table 1; File S2). Finally, the Methylophaga 1403 medium was replaced with the commercial Instant Ocean (IO) marine medium to mimic the original bioprocess. A third reactor was operated under the same conditions, this time with only strain NL23. The three reactors were run sequentially. Results from reactor 1 allowed us to adjust the conditions for reactor 2.

In reactor 1, the NO3− reduction rates and the denitrification rates (represented by the NOx reduction rates) ranged both between 1.6 and 5.1 mM h−1 (average 2.9 mM h−1) when operating with the Methylophaga 1403 medium at 0.5%, 1% and 2% NaCl (Table 4). At 2.75% NaCl, these two rates were lower (1.4 mM h−1) with a transient NO2− accumulation that peaked at 14 mM after 8 h (Table 4), suggesting a decrease in the denitrification performance. When the reactor was run with the IO medium, a 48-fold decrease occurred in the denitrification rates with accumulation of NO2− and a very low rate of its reduction afterwards (File S4). The gas production was absent reflecting low denitrifying activities. We suspected that the absence of the supplements (Wolf solution, solution T, and B12 vitamin) that were present in the Methylophaga 1403 medium could have caused this defect. After 12 days following re-addition of these supplements, the activities were partially restored (Table 4).

Table 4 Performance of the reactors.

	Gasa mL	Reduction rates mM h−1	NO2− Peakb	Proteinc	Specific denitrication rate (NOx) mM h−1 mg-prot−1	Ectoine µg/ support	
		NO3−	NOx					
Reactor 1	
Biofilm co-culture	
0.5%d	85	2.01	2.01	none	Nm		Nm	
1.0%d	95	1.62	1.62	none	Nm		Nm	
2.0%d	110	5.09	5.11	none	Nm		Nm	
2.75%d	100	1.40	1.44	13.8 (8)	Nm		Nm	
IO no sup	0	0.25	0.03	22.4 (96)e	Nm		Nm	
IO + sup	45	0.86	0.10	25.1 (24)f	227	0.43	9.8	
Reactor 2	
Biofilm co-culture	
0.5%	65	2.56	0.88	6.7 (5)	217	4.05	0.6	
1.0%	110	2.73	1.30	7.1 (4)	345	3.78	Nm	
2.0%	90	5.26	1.09	13.3 (3)	183	5.94	1.4	
2.75%	110	7.14	2.19	14.1 (2)	245	8.92	1.8	
IO + sup	105	5.20	0.94	12.9 (4)	391	1.99	17.4	
Reactor 3	
Strain NL23 biofilm mono-culture	
0%	150	5.12	5.04	none	Nm		Nm	
0.5%	140	2.74	2.74	none	49	56.2	Nm	
1.0%	140	2.57	2.68	none	79	34.0	Nm	
2.0%	90	2.70	0.53	12.9 (8)	97	5.5	Nm	
2.75%	25	3.23	0.11	LCg	34	3.3	Nm	
IO + sup	20	Nm	0.11	LCg	Nm	Nm	Nm	
Notes.

a Values are those when the gas production stabilized. Maximum theoretical N2 production is 132 mL based on that 11.77 mmoles of initial quantity of NO3− in the reactor (21.4 mM, 550 mL) would generate 5.9 mmole N2 gas (22.4 L mole−1).

b Transient NO2− accumulation: Peak concentration of NO2− (time in hours of the maximum peak measured).

c Estimated total protein in the reactor (suspended biomass and biofilm).

d Concentration of NaCl in the Methylophaga 1403 medium.

e More then 50% of NO2− was not reduced after 288 h.

f NO3− was reduced in less than 24 h with accumulation of NO2−. After 48 h latency, NO2− started to be consumed by the reactor and was completely reduced after 288 h.

g NO2− accumulation with low consumption (LC).

Nm, No measurement was carried out.

In reactor 2, the NO3− reduction rates were higher than those measured in reactor 1 when operating with the Methylophaga medium at 0.5%, 1% and 2% NaCl (Table 4), and ranged between 2.6 and 5.3 mM h−1 (average 3.5 mM h−1). These higher NO3− reduction rates could have had an impact on the reactor performance by generating transient NO2− accumulation that peaked between 7 to 13 mM after 2 to 4 h under these three conditions. Consequently of these transient accumulations, the denitrification rates in reactor 2 was lower than those in reactor 1 with values ranging between 0.9 and 1.3 mM h−1 (average 1.1 mM h−1). One reason of such differences between the two reactors is the higher proportion of strain NL23 in reactor 1, which ranged between 6.1 to 21.4% in these three conditions, compared to reactor 2 where these proportions ranged between 1.8 to 5.9% (Table 5). A higher proportion of the JAM1 strain would reduce NO3− faster and therefore generate NO2− faster than what strain NL23 could reduce at the same time. When reactor 2 was operated at 2.75% NaCl, the NO3− reduction rate and the denitrification rate increased by 36% and 101% respectively, in contrast to a 3.6-fold decrease of these rates in reactor 1 operating under the same conditions. A 5-fold decrease in proportion of strain NL23 (from 12.3% to 2.3%) in reactor 1 could explain these decreases, whereas this proportion stayed at similar level in reactor 2 (1.8% and 1.3%; Table 5). To avoid the detrimental effects observed in reactor 1, reactor 2 was run with the IO medium containing the supplements. The gas production, the NO3− reduction and denitrification rates, and the transient NO2− accumulation were similar than those observed in the other operating conditions (Table 4). Also, the proportions of strain NL23 in the suspended biomass were at similar levels (1.2 and 3.0%) than those measured in the previous operating conditions. In reactor 1, the proportion of strain NL23 was below 1% when operated with the IO medium (Table 5).

Table 5 Concentrations of the strains JAM1 and NL23 in the reactors determined by qPCR.

	Suspended biomass		Biofilm		Totala cp/reactor (x109)	Specific ratesb (µM-NOx h−1) per 109 gene copies	
	cp gene/mL (x107)	Proportion of NL23 %	cp/gene support (x107)	Proportion of NL23 %			
	JAM1	NL23	%	JAM1	NL23				
Reactor 1	
Biofilm co-culture	
0.5%	13.2 (28.0)	0.86 (0.90)	6.1	Nm	Nm				
1.0%	1.1 (1.6)	0.30 (0.11)	21.4	Nm	Nm				
2.0%	1.5 (0.5)	0.21 (0.15)	12.3	Nm	Nm				
2.75%	5.0 (1.3)	0.12 (0.10)	2.3	Nm	Nm				
IO no sup	42.9 (3.7)	0.34 (0.2)	0.78	Nm	Nm				
IO + sup	51.5 (54.6)	0.32 (0.30)	0.62	56.1	8.5	1.9	453	1.9	
Reactor 2	
Biofilm co-culture	
0.5%	60.9 (16.5)	2.7 (0.96)	4.2	92.8	11.4	10.9	621	1.42	
1.0%	46.2 (12.6)	2.9 (0.83)	5.9	Nm	Nm				
2.0%	23.3 (4.2)	0.42 (0.15)	1.8	Nm	Nm				
2.75%	14.4 (0.9)	0.19 (0.09)	1.3	262	12.8	4.7	796	2.75	
IO + sup	5.2 (4.6)	0.16 (0.08)	3.0	Nm	Nm				
IO no sup	10.5 (15.3)	0.13 (0.17)	1.2	182	8.1	4.3	554	1.62	
Reactor 3	
Strain NL23 biofilm mono-culture	
0%	NA	6.7 (3.2)		NA	Nm				
0.5%	NA	2.7 (1.4)		NA	23.3		76	36.1	
1.0%	NA	2.0 (0.86)		NA	15.4		51	52.7	
2.0%	NA	1.0 (0.59)		NA	8.2		27	19.6	
Notes.

Genes, napA for strain NL23; narG1 for strain JAM1. cp, copies. Values between parentheses are standard errors.

a Determined by the multiplication of the gene copies/mL of both strains by 550 mL (volume of the reactor) plus the multiplication of the gene copies per support of both strains by 260 supports.

b Specific rates were calculated by the division of the NOx reduction rates (Table 4) by the gene copies/reactor.

Nm, No measurement was carried out; NA, not applicable.

During the first four operating conditions, reactor 2 showed increases from 4.1 to 8.9 mM h−1 mg protein−1 in the specific denitrification rates (Table 4). No protein measurements were performed in reactor 1, except at the end of the running operations. When reactor 2 was operated with the IO medium, the specific denitrification rate decreased by 4.5 times. Once again, to avoid detrimental effect on the denitrifying activities, the supplements were removed one by one (except for trace metal elements) by operating the reactor for a few days between each removal. In all cases, complete denitrification occurred in less than 24 h, and gas production was constant throughout the operating conditions (Table 4).

In reactor 3 (run with only strain NL23), the NO3 − reduction rate and the denitrification rate were at 5.1 mM h−1 when operating with the Methylophaga 1403 medium at 0% NaCl. These rates dropped by 50% when operating at 0.5% and 1% (2.7 mM h−1; Table 4). At 2.0% NaCl, the NO3− reduction rate was still at 2.7 mM h−1, but NO2− accumulated and peaked at 13 mM after 8 h from which it was slowly reduced (File S4). Consequently, the denitrification rate was 5 times lower during this operating conditions compared to the previous conditions (2.7 to 0.53 mM h−1; Table 4). At 2.75% NaCl, the NO3− reduction rate was at similar level than the previous operating conditions, at 3.2 mM h−1 (Table 4). NO2− was still accumulated up to 16 mM after 6 h, but showed very low rate of reduction, where 11 mM NO2− was still present after 96 h (File S4). The denitrification rate was further down at 0.11 mM h−1. The reactor also showed low gas production. When the reactor was run with the IO medium, it showed the same behavior of the previous operating conditions with the denitrification rate at 0.11 mM h−1 as well as low gas production.

The levels of both strains were also determined in the biofilm of reactor 2 (Table 5). The proportions of strain NL23 were 3–4 times higher in the biofilm (4.3 to 10.9%) than in the suspended biomass, suggesting better environment for strain NL23 to grow. In reactor 3, with the increase of NaCl concentrations in the medium, the concentrations of strain NL23 decreased in both the suspended biomass and the biofilm (Table 5).

An indication of the level of specific denitrifying activities in the reactors was to divide the denitrification rates (mM-NOx h−1; Table 4) by the estimated total number of cells (strain NL23 and strain JAM1) in the reactor (suspended biomass and biofilm) as determined by qPCR (Table 5). These specific denitrification rates, translated per billion of cells (or gene copies), were at similar levels in reactor 2 during the different operating phases (Table 5). In reactor 3, the denitrification rates per billion cells were between 12 to 25 times higher than in reactor 2. It was at its lowest with medium at 2.0% NaCl, suggesting decreases in denitrification efficiency by strain NL23 cells.

Ectoine in the biofilm co-culture

We measured the level of the osmoprotectant ectoine in the biofilm of reactor 2 (Table 4). Increasing amount of ectoine was noticed with the increase of NaCl concentration in the reactor.

Biofilm structure

The colonization of surface by both strains was followed in the reactor with a flow chamber that was connected to the reactor and containing microscope slides (Fig. S1). Figures 3A and 3B showed uniform dispersed clusters of cells throughout the surface after one to five days of colonization. FISH assays showed that strain NL23 was lower in concentration than strain JAM1 in the biofilm (Figs. 3C and 3D), which concurs with the qPCR results. No specific pattern of distribution of both strains was noticed in the biofilm.

Figure 3 Biofilm visualization by fluorescence in situ hybridization.

Microscope slides were incubated in the flow cell chamber connected to the reactor 2 (biofilm co-culture) for 2–5 days to allow cell colonization. Cells were revealed by DAPI coloration or by FISH with specific probes targeting Methylophaga spp. and Hyphomicrobium spp., and examined by epifluorescence microscopy. Reactor operated with the Methylophaga 1403 medium at: 1% NaCl (A, B), at 0.5% NaCl (C), and with the IO medium (D). (A) Total cells revealed by DAPI coloration (200X). (B) H. nitrativorans NL23 (magenta) overlay with DAPI coloration (200X). (C–D) H. nitrativorans NL23 (orange) and M. nitratireducenticrescens JAM1 (green) (1000X). Scale bars represent 50 µm in (A) and (B), and 10 μm in (C) and (D).

Gene expression of denitrification genes in biofilm co-culture and mono-culture

The impact of the operating conditions in the reactors on the expression of denitrification genes of key reductases were assessed by measuring in the biofilm the levels of transcripts by RT-qPCR of both nar genes (narG1 and narG2) for strain JAM1, and napA and nirK for strain NL23. The transcript levels of narG1 did not change in the biofilm co-culture of reactor 2 operated at 0.5% and 2.75% NaCl, and with the IO medium (Table 6). For narG2, the transcript levels in the biofilm co-culture raised by about 2–3 times when the operating conditions in reactor 2 increased in NaCl concentration from 0.5% to 2.75% or when the medium was changed for the IO medium (Table 6).

Table 6 Changes in the transcript levels of selected denitrification genes determined by RT-qPCR.

	Strain NL23	Strain JAM1	
	napA	nirK	narG1	narG2	
Reactor 2	
0.5%	630 (347)a	478 (219)a	151 (18)a	54 (5)a	
2.75%	56 (18)b	224 (154)a,b	185 (33)a	88 (12)b	
IO	37 (6)b	77 (31)b	182 (100)a	140 (42)c	
Reactor 3	
0.5%	40 (19)b	180 (143)a,b	NA	NA	
2.0%	15 (7)b	286 (155)a,b	NA	NA	
Notes.

Values are expressed as gene copies per 100 copies of rpoB of respective strain. Values between parentheses are standard errors.

Superscript letter: For a specific gene, values with the same superscript letter between the biofilm samples are not significantly different (One-way analysis of variance, P < 0.05, Tukey’s multiple comparison test).

Results are from 4 to 9 RT-qPCR assays. NA: not applicable.

The napA transcript levels in the biofilm co-culture of reactor 2 had an 11-fold decrease when the operating conditions changed from 0.5% NaCl to 2.75% NaCl, or a 17-fold decrease when the medium was changed for the IO medium (Table 6). The napA transcript levels were similar in the biofilm mono-culture of reactor 3 operated with 0.5% and 2.0% NaCl. Finally, the nirK transcript levels showed no significant difference between the different operating conditions in reactors 2 and 3, except in reactor 2 operated with the IO medium with a 6-fold decrease compared to reactor 2 operated with 0.5% NaCl (Table 6).

The transcriptomic analysis

The impact of the operating conditions in gene expression of the pathways of denitrification and nitrogen assimilation were assessed by deriving the transcriptome of biofilm samples taken from reactors 2 and 3 operated under different conditions (Fig. 4). Except for napGH, no substantial changes were found in the relative transcript levels of the denitrification genes between the three operating conditions in reactor 2 (0.5% and 2.75% NaCl and IO) for both strains, and in the two operating conditions in reactor 3 (0.5% and 2.0% NaCl) for strain NL23 (Figs. 4A and 4B). Compared to reactor 2 operated with 0.5% NaCl, the relative transcript levels of napGH showed 3- to 4-fold increases in reactor 2 operated with 2.75% NaCl and with the IO medium, and a 9.3-fold increase in reactor 3 operated with 2% NaCl (Fig. 4B).

Figure 4 Changes in the relative transcript levels of genes involved in the denitrification and N-assimilation pathways, and in ectoine synthesis.

Total RNA was extracted from the biofilm samples taken from the reactors 2 and 3 operated at the different conditions. RNA samples were sequenced, and the reads associated to the denitrification genes (A and B), the N-assimilation genes (C) and ectoine genes (A) were transformed in transcripts per million (TPM). The TPM values of each gene operon were compared to those of the reactor 2, 0.5% NaCl, which was set to 1.0.

The type of medium (Methylophaga 1403 medium and IO medium) had a great impact on the N-assimilation pathways in both strains. Higher relative transcript levels were observed in both strains in reactors 2 and 3 (up to 105-fold increase in strain NL23) that were running with the IO medium for genes encoding ammonium transporters, and genes involved in the NO3− transformation in ammonium (Fig. 4C).

Finally, strain JAM1 showed 4- to-5-fold increases in the relative transcript levels of genes involved in ectoine synthesis in reactor 2 operated with 2.75% NaCl and with the IO medium, compared to the level found in the reactor 2 operated with 0.5% NaCl (Fig. 4A).

Discussion

The marine and methylotrophic environment that prevailed in the original denitrification system favored the establishment in the biofilm of two methylotrophic species, H. nitrativorans and M. nitratireducenticrescens directly involved in the denitrifying activities. In studying both species separately from the other microorganisms in the biofilm, we assessed what specific physiological characters of their own may have played important roles in the denitrifying activities in the original biofilm. One of these characters is the growth pattern. In pure cultures, M. nitratireducenticrescens strain JAM1 has shown to have a readiness to reduce NO3− with no lag phase for growth contrary to H. nitrativorans strain NL23, which has a 2–3 days lag phase before NO3− starts to be consumed and growth to occur. These results suggest that the denitrification pathway in strain NL23 takes 2–3 days to be induced under anoxic conditions. In strain JAM1 cells, this pathway is either induced quickly or it is constitutively expressed. Another character is the affinity towards NO3−. Compared to strain NL23, strain JAM1 has a higher µmax for growth and higher specific NO3− reduction rates, suggesting that strain JAM1 cells have a higher rate of NO3− processing (e.g., NO3− intake and reduction) than strain NL23 cells. This higher rate might be attributed to the different types of NO3− reductases used by these strains. Strain JAM1 possesses two Nar systems, which contribute both to the growth of strain JAM1 under anoxic conditions (Mauffrey, Martineau & Villemur, 2015), whereas strain NL23 has one periplasmic Nap-type NO3− reductases (Martineau, Mauffrey & Villemur, 2015). The Nar system is found associated with the membrane facing the cytoplasm. NO3− reduction by this system can generate proton motive force (PMF) and thus energy for the cells. Because of its location, the Nap system does not generate PMF upon NO3− reduction (Richardson et al., 2001). Furthermore, strain JAM1 possesses three NO3− transporters associated with the denitrification pathway. Differences in the regulation and the kinetics of the denitrification pathway could therefore confer an advantage of strain JAM1 for growth over strain NL23. These features are reflected in the higher abundance of strain JAM1 than strain NL23 that we found in the co-cultures as revealed by qPCR and FISH assays.

The planktonic co-cultures assays revealed the importance of strain NL23 to have its denitrification pathway stimulated to compete with strain JAM1 for growth. This stimulation resulted in better denitrification rates in planktonic co-cultures compared to those measured in strain NL23 pure cultures. In another study, Jeong & Kim (2019) showed that co-culturing the methylotroph Hyphomicrobium sp. strain NM3 with the methanotroph Methylocystis sp. strain M6 increased the methane-oxidation rate. In this system, strain NM3 would feed on inhibitory methyl-intermediates such as methanol as suggested by the authors.

The planktonic co-cultures did not reflect the true co-habitation of both strains under marine conditions that prevailed in the original denitrification system, since the planktonic co-cultures were carried out at low NaCl concentration to allow strain NL23 to grow. Such conditions, however, were met by carrying biofilm co-culture assays in the reactors. Acclimation of strain NL23 with increasing concentration of NaCl and then to the marine IO medium (medium used in the aquarium tanks that were treated by the original system) showed (i) sustained denitrifying activities, (ii) support colonization by strain NL23 (FISH assays) and (iii) higher proportions of strain NL23 in the biofilm co-culture compared to the suspended biomass.

With the increase of NaCl concentration in reactor 3 (run only with strain NL23), important decreases in denitrifying activities was noticed, specially when the NaCl concentration reached 2.0%. Martineau, Mauffrey & Villemur (2015) showed decreases in growth and denitrifying activities in planktonic pure cultures of strain NL23 cultured at 2% NaCl. However, reactor 3 operated in low NaCl concentrations (0.5% and 1%) had higher specific denitrification rates and higher denitrification rates per billions of cells that than those rates in reactor 2. Even at higher NaCl concentrations (2% and 2.75%), the specific denitrification rates were at similar level than those rates in reactor 2, and even much higher denitrification rates per billions of cells under these conditions. These differences in favor of strain NL23 can be explained by the lower level of biomass generated in this reactor (about four times less in protein; Table 4). Therefore, the presence of strain JAM1 did not contribute to better specific denitrifying activities in the reactor due to its growth prominence. However, its presence helped strain NL23 to achieve sustainable denitrifying activities at NaCl concentrations >1.0%.

Many factors could explain the stimulating effect of strain JAM1 on strain NL23 growth and activities in the biofilm such as the production of specific exopolysaccharides in the biofilm, quorum sensing and production of the osmoprotectant ectoine, as most of Methylophaga species can generate (Boden, 2019). Indeed, increase in ectoine production in the biofilm co-culture correlated with the increase in NaCl concentration in reactor 2. This higher level concurs with the higher relative transcript levels of the genes involved in ectoine production in the biofilm derived from the reactor 2 operated with 2.75% NaCl and IO compared to this reactor operated with 0.5% NaCl. Acclimation by strain NL23 of increases of NaCl concentration in the biofilm co-cultures could be related with an increase in ectoine production by strain JAM1. It is possible that the level of strain NL23 that can be reached in the biofilm is restricted by the level of ectoine generated by strain JAM1. Many bacteria have developed uptake transporter systems to accumulate osmoprotectant such as ectoine in their cells. For instance, Corynebacterium glutamicum possesses four secondary carriers for compatible solutes among which two, ProP and EctP, can uptake ectoine. Genes associated with these transporters belong respectively to the major facilitator superfamily and to the sodium/solute symporter superfamily (Peter et al., 1998). These types of transporters were found in the genome of strain NL23 (Martineau et al., 2013a).

Among the denitrification genes that we analyzed by RT-qPCR assays, only the NL23 napA presented a substantial change in its transcript levels, which were >10 times higher in the reactor 2 operated with 0.5% NaCl compared to the levels measured in the other operating conditions (2.75% NaCl and IO medium). However, such levels of napA transcripts were not observed in the biofilm mono-culture of reactor 3 operated with 0.5% NaCl, suggesting that strain JAM1 could have had a stimulating effect on the napA expression in the biofilm co-culture at this NaCl concentration. Wang et al. (2016) showed decrease of about 10 times in denitrification activities of a denitrifying granular sludge when the NaCl concentration increased gradually from 0% to 10% in synthetic wastewater. The same study observed increases in lethality and DNA leakage in these cultures when the NaCl concentrations were above 2%. Not many studies have reported the effect of salinity on the expression of denitrification genes. Gui, Chen & Ni (2017) observed decreases in the expression of the four denitrification genes (including napA) with the increase of NaCl concentration in the medium of Achromobacter sp. GAD-3 pure cultures.

Transcriptomic analysis showed 4- to 9-fold increases in the relative transcript levels of the NL23 napGH in reactors 2 and 3 operated with the 2.75% and with IO compared to the level found in reactor 2 operated with 0.5% NaCl. NapGH and NapC (encoded by the napEFDABC operon by strain NL23) have redundant function of transferring electrons to NapB across the membrane. It was proposed that NapC transfers electrons from the menaquinol, whereas NapGH do it from ubiquinol (Simon, 2011). The marine conditions may have favored the need of such proteins in the membrane of strain NL23.

We observed a drop of the denitrifying activities in reactor B2 with the passage from the Methylophaga 1403 medium to the IO medium. Besides IO to be commercial medium which could have unknown additives, differences between the two media are the presence of phosphate, ammonium and BisTris (buffer, and can bind some ions) in the Methylophaga 1403 medium that are absent in the IO medium. Ammonium can be a source of N that is readily assimilated in the glutamine pathway for biomolecule synthesis. In the IO medium, NO3− would have been transformed in ammonium before its assimilation, which requires energy from the cells. Indeed, we found higher levels of relative transcripts of genes associated with the N-assimilation pathways of both strains in the biofilms that were derived from reactors operated with IO.

Our results provide some explanations on how H. nitrativorans NL23 could have developed in the original biofilm. Because the original denitrification system operated under continuous mode, with constant concentration of NO3− in the affluent, strain NL23 would have had its denitrification pathway stimulated, thus capable to sustain growth, and not be outcompeted by strain JAM1. In such situation, strain NL23 would have benefited of the biofilm environment to acclimate under marine conditions, for instances by the availability of ectoine and the generation of NO2− by strain JAM1. This seems to concur with our previous study by Labbé et al. (2007), in which we showed by FISH that Hyphomicrobium spp. colonized surface much latter than Methylophaga spp. in the original denitrification system.

Conclusions

M. nitratireducenticrescens strain JAM1 has a higher rate of NO3− processing (e.g., NO3− intake and reduction) than H. nitrativorans strain NL23. Differences in the regulation and the kinetics of the denitrification pathway confer an advantage to strain JAM1 for growth over strain NL23. Better denitrification rates in planktonic co-cultures were obtained compared to strain NL23 planktonic mono-cultures but only in low salt medium. The reactors carrying the biofilm co-culture showed the acclimation of strain NL23 to increasing concentration of NaCl and then to the marine IO medium resulting in sustained denitrifying activities. The reactor which ran with only strain NL23 failed to achieve such activities under marine conditions. Although the presence of strain JAM1 did not contribute in better specific denitrifying activities in the reactor, its presence contributed for strain NL23 to carry sustainable denitrifying activities at NaCl concentrations >1.0%. Our results highlight some mechanisms of co-habitation from these two strains in achieving denitrifying activities, and can contribute in optimizing of denitrification systems treating saline/brackish waters.

Supplemental Information

Supplemental Information 1 Configuration of the reactor

Click here for additional data file.

Supplemental Information 2 Protocol for the measurement of the concentrations of NO3− and NO2−

Click here for additional data file.

Supplemental Information 3 Reactor operating procedures

Click here for additional data file.

Supplemental Information 4 Data for Figs. 1, 2 and 4

Click here for additional data file.

Supplemental Information 5 Data for Tables 4 and 5

Click here for additional data file.

Supplemental Information 6 Rt-qPCR results and statistical analysis

Click here for additional data file.

We thank Karla Vasquez and Sylvain Milot for their technical assistance.

Additional Information and Declarations

Competing Interests

Author Contributions

DNA Deposition

Data Availability

The authors declare there are no competing interests.

Alexandra Cucaita conceived and designed the experiments, performed the experiments, analyzed the data, prepared figures and/or tables, authored or reviewed drafts of the paper, and approved the final draft.

Marianne Piochon conceived and designed the experiments, performed the experiments, analyzed the data, authored or reviewed drafts of the paper, and approved the final draft.

Richard Villemur conceived and designed the experiments, analyzed the data, prepared figures and/or tables, authored or reviewed drafts of the paper, and approved the final draft.

The following information was supplied regarding the deposition of DNA sequences:

The transcriptomic sequences are available at GenBank: PRJNA744510, SRX11411255– SRX11411259.

The following information was supplied regarding data availability:

The complete protocol of nitrate and nitrite measurements; raw data for Figs. 1, 2 and 4 and Tables 4 and 5 ; the RT-qPCR results and the statistical analysis related to these results illustrated in Table 6 are available in the Supplementary Files.

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
