# Peer review of "Co-culturing Hyphomicrobium nitrativorans strain NL23 and Methylophaga nitratireducenticrescens strain JAM1 allows sustainable denitrifying activities under marine conditions"

_PeerJ, doi:10.7717/peerj.12424_

## Round 0.1 · original submission · Minor Revisions

We have obtained two high-quality review reports. Both reviewers praise your work, but they do suggest some areas where extra details must be provided. I believe addressing their requests will not be difficult.

Reviewer 1 ·

Basic reporting

The manuscript by Villemur et al. reports on the co-culture of two methylotrophs that together perform the denitrification pathway using methanol as carbon source. The pure culture profiles of the two microorganisms were compared with the co-culture profile in the presence of nitrate (planktonic and biofilm). Since JAM1 cannot reduce nitrite, the complete denitrification may only occur in the co-cultures, though the authors did not show this. The authors present evidence that co-culture enabled the growth of NL23 strain in the presence of NaCl (in a concentration that precluded growth of the pure culture).

The manuscript is well written, the Figures are explanatory and high quality. In addition, the experimental design to answer the hypotheses that were well made, and I agree with the majority of the conclusion drawn.
The background is provided in the introduction, but some references are missing relative to denitrification genes and denitrification of marine bacteria.

Experimental design

2. The amount of protein was determined for whole cells (without lysis) or after cell lysis?
3. The authors state that the study is on denitrification, however they have only monitored the expression of nar, nap and nir genes (Table 2). Thus, there is no indication that the pathway is active. Only that cultures can reduce nitrate and nitrate. Therefore, the title and also the text should be changed to take this into account and only mention nitrate and nitrite reduction.
7. Transcriptomic analysis: The primers used in this study are not described. How the values reported in Figure 5 were determined? Authors should report the original/raw data prior the calculations.

Validity of the findings

Thus, these are some of the major concerns:
1. The references to the denitrification pathway (1 to 3) should not be only of nitrate reductases but include references for each enzyme or for the study of the denitrification pathway. They could even refer manuscripts of the denitrification pathway of other marine bacteria (planktonic growth).
2. The amount of protein was determined for whole cells (without lysis) or after cell lysis?
3. The authors state that the study is on denitrification, however they have only monitored the expression of nar, nap and nir genes (Table 2). Thus, there is no indication that the pathway is active. Only that cultures can reduce nitrate and nitrate. Therefore, the title and also the text should be changed to take this into account and only mention nitrate and nitrite reduction.
4. Line 299 – 300: “Our results 300 showed that these co-cultures failed to perform complete denitrification.” What do the authors mean by complete denitrification? Nitrite reduction?
5. What the authors mean by denitrification rates? How were these measured?
6. What is the composition of the gas that is released?
7. Transcriptomic analysis: The primers used in this study are not described. How the values reported in Figure 5 were determined? Authors should report the original/raw data prior the calculations.
8. Both strain in co-culture (Biofilm or planktonic) are expressing nor and nos genes?

Additional comments

Other comments
1. Substitute “prescribed” in several sentences in the manuscript (Materials & Methods) with “different”. Indicate which concentrations were used in parenthesis.
2. Line 189: what is the meaning of virgin ?
3. Figure 1 should be transfer to Supplementary Materials.

Reviewer 2 ·

Basic reporting

Basic reporting
The article is generally well written, professionally structured and represents a self contained piece of work with relevant results. The introduction gives an excellent and concise background to the work, and referencing is appropriate throughout. However, it has minor English and grammatical mistakes throughout. Fixing these would make it easier to follow. As a few examples:

L26 “Planktonic pure culturing of both strains were performed”. Suggest changing to something like: “Pure cultures of both strains were grown with”. Similarly on L271.

L37: Change to “in contrast to”. Furthermore, this sentence has two different tenses for the two organisms i.e. “JAM1 had a” vs “NL23, has a”

L59 “We have been studying for many years”. I am not sure its needed to explicitly state that you have been working on it for many years needed as you succinctly and approriately cite the work in the rest of the paragraph. Suggest changing to something along the lines of “This study focuses on a biofilm derived from….

L60; Change ”that contains” to “containing”

Throughout: Transitory is used instead of transiently

L472 “ran only with” change to “run only with”

Figures: The presentation of the data in the figures and tables could be improved. The current layout makes it hard to interpret the results of the planktonic incubations and the biofilm study. In particular:

Figure 3: I would strongly suggest splitting this into 4 panels, the monocultures, and the two co-cultures. In its current format it is very hard to pull apart the different treatments. By separating this way, you could add other data such as the OD over time, and the ratios of the two organisms (if available).

Figure 4: Scale bars are missing

Tables 5 and 6: A figure detailing the operation of the three reactor runs is essential to interpret the results. Such a figure could incorporate nitrate and nitrite concentrations and when the media was changed, the proportions of the two organisms and the periods where rates were calculated. In essence this would be a visual representation of table 5 and 6, which contain a lot of information that is hard to compare. The tables could then be combined showing just the rates. Furthermore, please be more specific as to what “not measured” means… was there nothing to be measured or did no measurement happen?

Raw data: Where have the results from the RNA sequencing been deposited?

Experimental design

The manuscript investigates the nitrate uptake/reduction capacities of two methylotrophic denitrifiers (one which misses a nitrite reductase and one with a full denitrifying pathway) which, prior to isolation, dominated a biofilm in a reactor that treated a seawater aquarium. The authors overall aim was to enhance our understanding of how these two taxas co-existence might enhance denitrification systems treating saline/brackish waters. The main results of the study are that strain JAM1 is a better competitor for nitrate; it had a higher µmax, a lower ks and a better µmax/ks ratio, and it appeared to switch to nitrate respiration quicker. Furthermore, strain NL23 appeared to be sensitive to increasing salinity. As such, JAM1 dominated co-cultures both in planktonic and biofilm set ups and the authors conclude that its presence is “essential” for the operation of NL23 at NaCl concentrations more than 1%. There are some issues that need to be addressed in regards to the experimental design.

1) The research question is currently not well defined/consistent throughout the manuscript. The introduction introduces the hypothesis that “that a tight relationship has developed between H. nitrativorans strain NL23 and M. nitratireducenticrescens strain JAM1 to achieve denitrification in the original biofilm”. I would argue that this is not really a testable hypothesis and does not reflect the rest of the manuscript, it is not addressed explicitly in the conclusions. Furthermore the discussion introduces the hypothesis that “JAM1 has to be present for strain NL23 to survive under marine conditions”. I would suggest that the authors are more specific about the research questions that their experiments were designed to address at the end of the introduction.

2) There are a number of details missing from the methods which made the results difficult to interpret. I think the most important omission is details regarding the planktonic culture experiments that inform figure 2. How many vials were set up per treatment? How often were they sampled, and for what?

3) One of the few methodological details given is that the cultures were incubated for 1 – 7 days without shaking. Considering the biofilm capacity of both of these organisms how can you rule out that biofilms formed in the vials? The presence of a mixture of biofilm and planktonic cells in the vials could have a considerable impact on the apparent affinities due to diffusion limitation.

4) For all rates, please insert a table detailing the results of the linear regression used to calculate them with significance and error values. Similarly, for results from the bioreactors where estimates have been made based on two measurements i.e. denitrification rate per mg protein, what is the propogated error?

5) Other details which should be added are:

L 142. Where did the material come from for the pre-cultures? Here it would be good to cite again the initial study describing their isolation and submission to relevant culture collections.

173. How was air added with a peristaltic pump?
How did you confirm anoxic conditions in the bioreactor, and are you sure that they were maintained. For example, how did you collect supports from the bioreactor without introducing O2? How did you change the media and ensure that the media added was anoxic?

What were the timings for sampling and when exactly was media exchanged in the bioreactors (see my comment about adding in a figure detailing the bioreactor performance and operation).

How was reactor 3 made anoxic from the start?

Considering the results of the planktonic experiments, why did you choose to inoculate the bioreactors with similar biomass of the two organisms?

Validity of the findings

Validity of the findings:

The major finding in the manuscript is that JAM1 is essential for NL23 to “operate” under higher salinity conditions. Currently this finding is not strongly supported. In the results (L355 onwards), it is stated that denitrification still occurred in reactor 3 (NL23 only), although at much lower rates as salinity was increased and before switching to IO media, therefore JAM1 is not “essential”. From the information given in table 5, it does appear that at 2.75 % NaCl, the co-culture had higher NO3- reduction and specific denitrification rates and I would suggest reformulating your discussion to reflect this more explicitly.

Secondly, I find your discussion about the differences in nitrate uptake/reduction from L443 a very convincing explanation for why JAM1 is a better competitor for nitrate and seems to outcompete NK23 in the planktonic experiments. Taking into account that NL23 also seems sensitive to salinity, I think an important question is therefore: How did these two organisms c-exist at high abundances in the original aquarium bioreactor film? Surely if NL23 is such a poor competitor for nitrate in terms of growth and affinity (if all other conditions are equal), there must have been another factor that allowed it to persist? If you do not address this, then you do not address your original intent to enhance understanding of how denitrification systems treating saline/brackish waters can be optimized.

Minor issues should also be addressed to make the interpretation of the results clearer:
What would have caused the initial difference in reactor 1 and 2 that lead to the higher rates and accumulation of nitrite?

How much gas would you expect to be produced based on the biomass and denitrification rate? Can you really guarantee that your system wasn’t leaking? What is the relevance of this result?

L433. I would suggest rephrasing this line. Why would an equilibrium have to be reached? Or did you assume an equilibrium was reached and here you assess how the physiological characteristics lead to the establishment of a biofilm with 2 species?

I do not think dynamics/dynamism is an appropriate term, their nitrate utilization characteristics? Alternatively, you must define what you mean by dynamism in the results and discussion.

What do you suggest that NL23 does in the lag phase? If it is not reducing nitrate as an electron acceptor, how does it avoid becoming “trapped in anoxia?”.

L438 I think that it means that its quickly induced, your results say nothing about its regulation.

---

## Round 0.2 · accepted · Accept

I am happy with the changes introduced.

Reviewer 1 ·

Basic reporting

The manuscript was modified according to the reviewers suggestions (when possible). It is in a state to be published.
However, the reviewer suggests to correct all values with the erros in the manuscript:
an example: 0.428 ± 0.019 should be 0.43 ± 0.02.

Experimental design

nothing to report

Validity of the findings

nothing to report